# TLR7 Promotes Acute Inflammatory-Driven Lung Dysfunction in Influenza-Infected Mice but Prevents Late Airway Hyperresponsiveness

**DOI:** 10.3390/ijms252413699

**Published:** 2024-12-21

**Authors:** Mark A. Miles, Stella Liong, Felicia Liong, Gemma S. Trollope, Hao Wang, Robert D. Brooks, Steven Bozinovski, John J. O’Leary, Doug A. Brooks, Stavros Selemidis

**Affiliations:** 1Centre for Respiratory Science and Health, School of Health and Biomedical Sciences, RMIT University, Bundoora, VIC 3083, Australia; mark.miles@rmit.edu.au (M.A.M.); stella.liong@rmit.edu.au (S.L.); felicia.liong@rmit.edu.au (F.L.); gemma.trollope@rmit.edu.au (G.S.T.); hao.wang@rmit.edu.au (H.W.); steven.bozinovski@rmit.edu.au (S.B.); 2Clinical and Health Sciences, University of South Australia, Adelaide, SA 5001, Australia; robbrooksbd@gmail.com (R.D.B.); doug.brooks@unisa.edu.au (D.A.B.); 3Discipline of Histopathology, School of Medicine, Trinity Translational Medicine Institute (TTMI), Trinity College Dublin, D08 XW7X Dublin, Ireland; 4Sir Patrick Dun’s Laboratory, Central Pathology Laboratory, St James’s Hospital, D08 XW7X Dublin, Ireland

**Keywords:** toll-like receptor 7, influenza A, lung function, inflammation, infection

## Abstract

Severe lower respiratory tract disease following influenza A virus (IAV) infection is characterized by excessive inflammation and lung tissue damage, and this can impair lung function. The effect of toll-like receptor 7 (TLR7), which detects viral RNA to initiate antiviral and proinflammatory responses to IAV, on lung function during peak infection and in the resolution phase is not fully understood. Using wild-type (WT) C57BL/6 and TLR7 knockout (TLR7 KO) mice, we found that IAV infection induced airway dysfunction in both genotypes, although in TLR7 KO mice, this dysfunction manifested later, did not affect lung tissue elastance and damping, and was associated with a different immune phenotype. A positive correlation was found between lung dysfunction and the infiltration of neutrophils and Ly6C^lo^ patrolling monocytes at day 7 post-infection. Conversely, in TLR7 KO mice, eosinophil and CD8+ cytotoxic T cells were associated with airway hyperactivity at day 14. IL-5 expression was higher in the airways of IAV-infected TLR7 KO mice, suggesting an enhanced Th2 response due to TLR7 deficiency. This study highlights an underappreciated duality of TLR7 in IAV disease: promoting inflammation-driven lung dysfunction during the acute infection but suppressing eosinophilic and CD8+ T cell-dependent hyperresponsiveness during disease resolution.

## 1. Introduction

Each year, approximately 1 billion people are infected with influenza A virus (IAV), with up to 650,000 cases resulting in death [1]. Severe respiratory disease often manifests in lower respiratory tract infection, leading to serious pulmonary complications that can impair lung function, cause conditions like bronchiolitis, and, in vulnerable individuals, even result in death [2]. Pandemic IAV strains such as H1N1 cause significantly more hospitalizations and deaths compared to seasonal strains, particularly affecting the young [3]. These patients often experience excessive pulmonary inflammation due to a “cytokine storm”, which provokes lung immunopathology and acute respiratory distress [4,5,6]. Indeed, exacerbated cytokine responses enhance disease pathology in IAV mouse models [7,8,9]. Additionally, IAV-induced inflammation impairs lung function, increasing the susceptibility to allergen sensitization and airway hyperresponsiveness [10,11,12,13]. Therefore, the hyperactivation of host inflammatory responses by IAV can play a critical role in driving both acute and chronic respiratory complications following infection.

Endosomal toll-like receptor 7 (TLR7) plays a crucial role in the immune response to IAV infection [14], sensing viral single-stranded RNA (ssRNA) following internalization of the virus into the cell by endocytosis. TLR7 activation triggers the transcription of type I interferon (IFN) via interferon regulatory factors and/or proinflammatory cytokines via nuclear factor kappa-light-chain-enhancer of activated B cells (NFκB) to promote innate and adaptive immune responses [15]. Despite this, TLR7-deficient mice can still manage IAV infection, although the innate and CD4+ T helper responses and production of virus-specific antibodies are somewhat impaired [16,17,18,19,20]. The ability to control an IAV infection, despite the absence of TLR7, is in part due to compensatory antiviral signaling, such as those involving retinoic acid-inducible gene I (RIG-I)-like receptors that sense viral RNA in the cytosol and promote inflammation to limit viral replication. Therefore, while TLR7 is not strictly essential for managing acute IAV infection, its high expression in B cells underscores its important role in humoral immunity and antibody production [19].

Early or prophylactic stimulation of TLR7 using the TLR7 agonist imiquimod has been shown to offer protection against IAV [21,22]. This protection is presumably due to heightened antiviral IFN responses during the initial stages of infection, which help block viral replication. However, unrestrained TLR7 activation can lead to hyperinflammatory responses, potentially causing host-mediated immunopathology following IAV infection. Similar hyperinflammatory responses have been observed with other ssRNA viruses like respiratory syncytial virus (RSV) and severe acute respiratory syndrome coronavirus- 2 (SARS-CoV-2) [23,24], as well as in diseases such as chronic obstructive pulmonary disease, sepsis, autoimmunity, and cancer [25,26,27,28]. Indeed, IAV-infected mice treated with the TLR7 antagonist IRS661 at 4 days post-infection (dpi) displayed reduced lung pathology and had improved survival [29]. IRS661 treatment did not affect viral load but had reduced pulmonary cytokine release and innate cell recruitment, emphasizing the role of a TLR7-mediated cytokine storm in exacerbating IAV disease. These studies suggest a potential dual role of TLR7 during IAV infection. We postulated that the activation of TLR7 at the onset of infection can boost early antiviral IFN responses for protection against the virus while inhibiting TLR7 once an infection is established could suppress the cytokine storm and inflammatory lung damage.

Given this context, it remains to be proven how TLR7-related inflammation affects lung function following IAV infection. To assess the contribution of TLR7 to IAV-induced lung dysfunction, we employed a global TLR7 knockout (TLR7 KO) mouse model infected with the highly pathogenic PR8 strain (H1N1). Analyses were performed on days 7 and 14 post-infection corresponding to peak inflammation and recovery phases, respectively. Our findings reveal two distinct functional roles of TLR7 in lung function during infection: (1) promoting acute inflammatory-driven lung dysfunction and (2) mitigating the onset of Th2-associated airway hyperresponsiveness.

## 2. Results

### 2.1. Slight Delay in Acute Bodyweight Loss and Reduced Viral Load in TLR7-Deficient Mice

Mice were inoculated intranasally with a sublethal dose of PR8 or PBS as a control, and bodyweight loss was monitored as an indicator of infection-induced morbidity. Compared to the uninfected group, infected WT mice exhibited significant bodyweight loss beginning at 5 dpi. Infected TLR7 KO mice, however, exhibited significant bodyweight loss from day 7 onwards (Figure 1A). There was no significant difference between infected WT and TLR7 KO mice, although it should be noted that uninfected WT gained more weight across the experiment compared to TLR7 KO control mice. Both genotypes experienced a similar peak bodyweight loss of approximately 10–15% between 8 and 9 dpi, after which they began to regain weight, surpassing baseline levels by day 14. Notably, infected TLR7 KO mice maintained a significant weight difference compared to the PBS control group until 12 dpi, which was 2 days longer than observed in WT mice, suggesting a delay in overall kinetics. Viral transcripts were detected in the lungs at 7 dpi, which represents peak infection, but were undetectable by day 14, suggesting that the virus was largely cleared by this time (Figure 1B). Additionally, the viral load at 7 dpi was significantly lower in TLR7 KO mice.

These findings indicate that sublethal PR8 infection leads to comparable levels of acute bodyweight loss in both genotypes, although TLR7 KO mice experienced a slightly delayed onset and recovery. Both genotypes eventually recovered from virus-induced bodyweight loss and had cleared the virus from their lungs by 14 dpi, indicating they had overcome the peak of the acute disease by this timepoint.

### 2.2. IAV-Induced Dysfunction of Large Airways Is Delayed in TLR7-Deficient Mice but Lung Tissue Function Is Spared

To assess differences in IAV-induced lung dysfunction, we monitored lung mechanics on days 7 and 14, both at baseline and following methacholine (MCh) challenge, as this mimics the bronchoconstriction observed in asthma. These timepoints coincide with peak inflammation and viral clearance [30], with all mice showing recovery of bodyweight loss and viral clearance by day 14. Infected WT mice displayed significant increases in baseline total respiratory system resistance (Rrs) and total respiratory system elastance (Ers) at 7 dpi, which returned to uninfected (baseline) levels by day 14 (Figure 2A). In contrast, TLR7 KO mice showed a reduced baseline increase in Rrs and Ers compared to WT mice at 7 dpi with a slight upward trend by day 14 (Rrs *p* = 0.2119, Ers *p* = 0.2095). There were no significant changes in the baseline resistance of the conducting airways (Rn) with infection in either genotype. Upon MCh challenge, infected WT mice showed a significant increase in Rrs at 7 dpi, indicating hyperresponsiveness, which returned to uninfected levels by day 14 (Figure 2B,C). A similar trend was observed for Ers. In TLR7 KO mice, these acute MCh-induced increases in Rrs values were suppressed but appeared to rise at 14 dpi, suggesting delayed lung hyperreactivity but preserved lung elasticity. According to a relative analysis, both WT and TLR7 KO mice displayed increased Rn at 7 dpi, with TLR7 KO maintaining higher Rn at 14 dpi compared to WT mice, which trended lower (*p* = 0.0895), and likely accounted for the higher Rrs values in these mice. This indicates that TLR7 deficiency delays acute lung resistance and hyperreactivity during IAV infection, while hyperreactivity of the conducting airways persists.

Tissue damping (G) and tissue elastance (H), which indicate how energy dissipates or is conserved in the alveoli were measured to assess the degree of alveolar dysfunction within the lung tissue. Infected WT mice displayed significant increases in baseline G and H resistance at 7 dpi (Figure 3A), indicating impaired expansion of the lung tissue. By day 14, these values had returned to levels seen in uninfected mice. The baseline increase in G was blunted in TLR7 KO mice at 7 dpi (*p* = 0.0773). However, by day 14, baseline H values in TLR7 KO mice were significantly elevated compared to uninfected mice, indicating increased alveolar stiffness, and were even higher than those in WT mice at the same timepoint. Unlike in WT mice, TLR7 KO mice did not display an increase in G and H following MCh challenge at 7 dpi (Figure 3B,C) or develop resistance at 14 dpi. These data demonstrate that TLR7 contributes to both baseline and MCh-induced dysfunction of the lung tissue during PR8 infection, whereas TLR7 deficiency protects against such dysfunction of the alveoli, unlike the situation in the larger airways.

### 2.3. TLR7 Deficiency Reduces IAV-Induced Pulmonary Immune Cell Recruitment at 7 dpi but Promotes Monocytosis, Neutrophilia, and Eosinophilia at 14 dpi

The data thus far indicate that acute lung dysfunction at 7 dpi occurs via TLR7, marking the peak morbidity. Analysis of the immune cell composition in the lungs of WT mice at 7 dpi revealed that PR8 infection led to an increased influx of various innate inflammatory cells, including macrophages, neutrophils, Ly6C^lo^ and Ly6C^hi^ monocytes, and both classic (cDC) and plasmacytoid (pDC) dendritic cells (Figure 4A). The lung infiltration of these cell types was significantly suppressed in TLR7 KO mice, though dendritic cell numbers remained elevated relative to the uninfected control group. At 14 dpi, macrophage and DC subpopulations remained significantly elevated in infected WT mice, while neutrophil and Ly6C^lo^ and Ly6C^hi^ monocyte numbers returned to uninfected levels. Both cDC and pDC subtypes also remained higher in TLR7 KO mice at this timepoint, similar to WT mice. Despite no alterations at 7 dpi, the composition of TLR7 KO lungs contained more neutrophils and Ly6C^lo^ and Ly6C^hi^ monocytes at 14 dpi compared to the uninfected controls, and this was significantly higher than in WT mice. This suggests a delayed innate inflammatory response to PR8 infection in the absence of TLR7. Moreover, while eosinophil populations in WT mice remained unchanged throughout the infection, TLR7 KO mice showed a significant increase in eosinophils in the lungs at 14 dpi. This suggests that TLR7 plays a role in suppressing eosinophilia during IAV infection.

Analysis of T cell populations in the lungs (Figure 4B) revealed similar increases in T regulatory cells (Treg) and CD8+ cytotoxic T cells following infection in both genotypes at 7 dpi, while significantly more CD4+ helper T cells were found in TLR7 KO mice. The number of CD4+ or CD8+ T cells stained for activation marker CD69 was significantly lower in TLR7 KO mice, suggesting reduced numbers of activated T cells in these mice. At 14 dpi, WT-infected mice had a significantly higher proportion of CD4+ helper T cells and Tregs compared to TLR7 KO mice, although the frequency of activated CD4+ helper T cells was similar between genotypes. Both WT and TLR7 KO mice displayed similar increases in CD8+ cytotoxic T cells in response to infection; however, the proportion of these cells expressing CD69 was significantly lower in TLR7 KO mice. Analysis of CD8:CD4 T cell ratios revealed a predominance in CD8+ cytotoxic T cells in the lungs of WT mice at 7 dpi, whereas TLR7 KO mice displayed a significant shift towards a higher CD8:CD4 T cell ratio by 14 dpi.

These data indicate that the influx of inflammatory myeloid cells to the lungs was delayed in TLR7-deficient mice, which was associated with an increase in eosinophil recruitment. The recruitment of CD8+ cytotoxic T cells was not compromised, although these cells exhibited a reduced activation status.

### 2.4. The IAV-Induced Lung Dysfunction Phenotype Changes with TLR7 Deficiency

To identify the cell types contributing to IAV-induced lung dysfunction, we performed a linear regression analysis to examine the relationship between immune cell populations in the lungs and the respiratory system resistance across infections in all mice (Table 1). At baseline, on 7 dpi, we found a significant positive correlation between Rrs and lung neutrophils, Ly6C^lo^ patrolling monocytes, and the CD8:CD4 T cell ratio (Figure 5A and Appendix A). Although cDC (*p* = 0.0894) and pDC (*p* = 0.055) populations showed a positive trend, their correlation with Rrs was not significant. Following a MCh challenge at 7 dpi, mice with significantly increased Rrs hyperresponsiveness also had elevated levels of lung neutrophils and Ly6C^lo^ patrolling monocytes (Figure 5B and Appendix A). However, no significant relationship was found for the lung CD8:CD4 T cell ratio in this context. These findings suggest that neutrophils and Ly6C^lo^ patrolling monocytes contribute to both the baseline and MCh-induced hyperresponsiveness at 7 dpi. A distinct relationship emerged at 14 dpi. At baseline, on 14 dpi, a significant positive correlation was identified between Rrs and the levels of lung Ly6C^hi^ inflammatory monocytes, eosinophils, and the CD8:CD4 T cell ratio (Figure 5C and Appendix A). A negative correlation was observed for NK cells and Tregs. Following the MCh challenge at 14 dpi, mice experiencing significant Rrs hyperresponsiveness also contained high levels of lung Ly6C^hi^ inflammatory monocytes, eosinophils, CD8+ T cells, and CD8:CD4 T cell ratios (Figure 5D and Appendix A). These mice also had significantly lower numbers of NK cells. Similar results were observed when comparing lung immune cell infiltration against Rn values (Appendix A).

These data highlight that lung dysfunction at 7 dpi is consistently associated with elevated frequencies of neutrophils and Ly6C^lo^ patrolling monocytes in the lungs, and this was mainly observed in WT mice. In contrast, at 14 dpi, where lung hyperreactivity was noted in TLR7 KO mice, there was a strong association with higher frequencies of Ly6C^hi^ inflammatory monocytes and eosinophils. Also, CD8+ cytotoxic T cells correlated with Rrs only at 14 dpi following MCh challenge. These findings indicate that TLR7 deficiency alters the phenotype of large airway dysfunction and may suggest a shift towards a Th2-skewed immune response.

### 2.5. TLR7 Deficiency Suppresses the Cytokine Storm but Enhances Th2 Signaling

To determine whether the observed alterations in lung dysfunction were due to specific differences in inflammatory signaling during infection, cytokine levels in the bronchoalveolar lavage fluid (BALF) were analyzed to assess inflammatory protein secretion into the airways. After 7 days of infection, the levels of Th1 cytokines IFNγ, IL-6, and TNFα were elevated in the BALF, but these levels were significantly reduced in TLR7 KO mice (Figure 6A). Similarly, the chemokines CCL2 (attracting monocytes), CXCL2 (attracting neutrophils), CCL5 (attracting T cells), CXCL10 (attracting activated T cells, eosinophils), and CCL11 (eosinophils) were elevated during infection with a significant reduction in TLR7 KO mice (Figure 6B). This reduction corresponds with lower total cytokine levels and fewer macrophages, monocytes, neutrophils, and activated CD8+ T cells in the lungs of TLR7 KO at this timepoint. Interestingly, IL-5 expression (but not IL-4) increased in both genotypes by 7 dpi, although it was significantly higher in TLR7 KO mice (Figure 6C), indicating a more enhanced Th2 response in these mice. The Th1:Th2 ratio, calculated from IFNγ:IL-4 or IFNγ:IL-5, indicated a predominant Th1 response in the BALF of both genotypes at 7 dpi. However, this Th1 response was significantly lower in TLR7 KO mice, indicating reduced overall inflammation. At 14 dpi, IFNγ levels remained elevated in the BALF of WT mice but not in TLR7 KO mice, consistent with the magnitude of activated CD8+ T cells in WT mice and suggesting a persistent inflammatory response. In contrast, CXCL2, CXCL10, and CCL11 levels were higher in TLR7 KO mice at 14 dpi, while they remained unchanged in WT mice, consistent with the higher numbers of neutrophils, monocytes, and eosinophils in the lungs of TLR7 KO mice. The remaining proteins tested did not show differences between infected and uninfected groups at this later timepoint. This suggests that the overall cytokine response had largely been resolved in the airways of WT mice by day 14, except for IFNγ, while certain chemokine levels continued to be elevated in TLR7 KO mice.

## 3. Discussion

In this study, we aimed to better understand how host inflammatory responses during IAV infection led to lung dysfunction. Our findings demonstrate that infection with the PR8 strain of IAV induces extensive lung inflammation, immune cell recruitment, and functional impairments in both large conducting airways and in the lung tissue after 7 days. This includes increased baseline airway resistance and heightened hyperresponsiveness to MCh. These results are consistent with previous studies describing acute lung dysfunction following PR8 infection [10,31,32]. Importantly, we found that TLR7 deficiency lessened lung inflammation and suppressed the onset of lung hyperresponsiveness during acute infection. We also performed measurements at day 14 to assess if lung dysfunction persisted after the resolution of the infection. At this point, airway inflammatory cytokine levels had largely returned to uninfected levels in both TLR7 KO and WT mice. However, TLR7 KO mice continued to exhibit hyperresponsiveness in the large airways, while WT mice had a resolution of lung dysfunction. Our correlation analyses identified distinct pulmonary immune cell profiles and lung dysfunction between TLR7-proficient and TLR7-deficient mice, arguing that the pathogenic mechanisms driving airway dysfunction differed between these mice. This study is the first to report the pathogenic role of TLR7 in lung function during both acute and resolved phases of IAV infection.

The dynamic recruitment of various inflammatory cells to the lungs during peak infection is important for clearing virally infected cells, but the magnitude of this response may also compromise respiratory function. At 7 dpi, both baseline and MCh-induced respiratory dysfunction were observed in WT mice and were strongly correlated with lung neutrophil and Ly6C^lo^ patrolling monocyte populations. The reduced Th1 inflammatory signaling in TLR7 KO mice at this timepoint further implicates TLR7-mediated inflammation as a key driver of pathology in this process. Neutrophils and monocytes are readily recruited to the lungs during the initial stages of IAV infection, where they secrete high amounts of inflammatory cytokines. Such early responses are important for limiting viral replication [33]; however, excessive and persistent inflammation mediated by neutrophils and monocytes can cause damage to lung tissue and exacerbate pathology [34,35,36,37]. The tissue remodeling mediated by these cells can adversely impact airway structure and function, leading to lung function decline and increased airway hyperresponsiveness [38,39]. High numbers of classical and CD206-ve monocyte subtypes have been detected in the sputum of patients with neutrophilic asthma, highlighting their role in chronic airway disease [40]. Furthermore, increased levels of CCL2, a strong monocyte chemoattractant, were observed in the BALF of WT mice in our study, mirroring findings in asthmatic patients where elevated CCL2 levels were noted compared to healthy controls [41]. These studies emphasize the contribution of these cell types towards airway disease.

Dysregulation of the crosstalk between neutrophils and monocytes can alter inflammation and contribute to disease progression [42]. Our study implies that this dysregulation plays a role in driving lung dysfunction during peak IAV infection. A TLR7-dependent relationship between Ly6C^lo^ patrolling monocytes and neutrophils has been reported previously [43,44]. Activation of TLR7 in Ly6C^lo^ patrolling monocytes increases their retention on the endothelium, facilitating the recruitment of neutrophils that are involved in necrosis and the removal of dead endothelial cells [43]. Inflammation prolongs this interaction, which enhances neutrophil-mediated ROS production, cytokine release, and tissue damage [45]. In the context of IAV infection, elevated numbers of monocytes, neutrophils, and T cells, along with increased TLR7 transcripts, have been observed in the aorta of IAV-infected pregnant mice. This results in aortic inflammation and endothelial dysfunction [46]. Administration of the TLR7 agonist RESIQUIMOD also promoted rapid monocyte egress from bone marrow and their differentiation to vascular Ly6C^lo^ monocytes and macrophages in lung tissue, resulting in reduced IAV morbidity [22]. Given that Ly6C^lo^ patrolling monocytes express high levels of TLR7 [47], persistent TLR7 stimulation during infections with high viral load could perpetuate inflammation driven by the monocyte–neutrophil interaction. Rappe, Finsterbusch, Crotta, Mack, Priestnall, and Wack [29] showed that TLR7 enhanced the production of neutrophil chemoattractants Mip-1α and Mip-2 by monocytes following IAV infection in IFNAR1 knockout mice, exacerbating immunopathology. Another potential mechanism involves the release of damage-associated molecular patterns following neutrophil-mediated tissue damage, which was shown to activate the NLRP3 inflammasome in alveolar macrophages [48]. As TLR7 promotes the NFκB-mediated transcription of immature IL-1β and IL-18 in response to viral RNA, neutrophils could propagate mature inflammasome activation and exacerbate pathological inflammation via this pathway [9]. As such, we hypothesize that inflammation resulting from monocyte–neutrophil interactions is lower in TLR7-deficient cells following IAV infection, presumably due to reduced monocyte activation, which protects mice from lung dysfunction. Further work is needed to investigate the TLR7–monocyte–neutrophil interaction in IAV infection and to determine whether similar interactions occur outside the endothelium in the lung.

TLR7 deficiency was able to block the monocyte and neutrophil association with lung dysfunction during acute disease but did not prevent the later onset of airway dysfunction, suggesting other infiltrating cell types may be involved in this process. Indeed, the phenotype of airway hyperreactivity exhibited by TLR7 KO mice at 14 dpi was instead largely associated with lung eosinophil, Ly6C^hi^ inflammatory monocytes, and CD8+ T cell populations. In contrast, WT mice did not exhibit eosinophilia during the infection, implying that TLR7 loss altered the immune response to enable greater Th2 signaling. This shift to Th2 dominance might have contributed to the IAV-induced lung dysfunction in TLR7-deficient mice after the acute phase of the infection. Previous studies have highlighted that aberrant Th2 and Th17 immune signaling can lead to lung immunopathology and dysfunction in some models of IAV infection [10,31]. Similar immunological responses to IAV have been associated with exacerbated airway hyperresponsiveness to subsequent antigen exposure [13,49,50]. The ability of TLR7 signaling to promote an inflammatory response favoring Th1 immunity helps counteract the Th2-skewed imbalance observed in allergic or asthma models [51,52,53,54]. In the context of IAV infection, TLR7 deficiency promoted a heightened Th2 response attributed to the increased accumulation of myeloid-derived suppressor cells in the lung [55]. Our data are consistent with this, as IAV-induced lung eosinophilia and the elevated secretion of the Th2 cytokine IL-5 were observed in TLR7 KO. Prior research showed that lung-infiltrating Ly6C^hi^ monocytes can enhance eosinophil recruitment, linking these cell types to allergic inflammation [56,57]. Moreover, TLR7 has also been implicated in regulating eosinophil-driven allergic airway inflammation following infection with other RNA viruses such as rhinovirus or Sendai virus [58,59]. At 14 dpi, TLR7 KO mice also had elevated levels of eotaxin (CCL11) and CXCL10, both of which are chemokines for eosinophil recruitment. CXCL10 acts on the CXCR3 receptor expressed on activated T lymphocytes and eosinophils [60], and has been implicated in promoting IAV-induced lung pathology [61,62]. CXCL10 upregulation in eosinophilic asthmatic mice [63] has been linked to increased airway hyperreactivity, eosinophilia, and CD8+ T cell recruitment, all of which were evident in TLR7 KO mice at 14 dpi. Accompanying the increase in eosinophils in TLR7 KO mice was a reduction in NK cell frequency. NK cells have previously been shown to suppress airway hyperreactivity in ovalbumin-sensitized mice [64] and to negatively regulate eosinophilic inflammation [65]. The decrease in NK cells observed in the hyperresponsive TLR7 KO mice could therefore contribute to the increased eosinophilia and resultant airway hyperreactivity. Thus, the absence of TLR7 may enhance the Th2 signal in the lungs via a mechanism that suppresses a persistent NK cell response and enhances eosinophilic pathology. Further research could also consider the effect of TLR7 on other pathways involved in the amplification of the Th2 response, such as IL-13, IL-31, and IL-33 cytokines, vitamin D deficiency, and changes in the lung microbiome [66,67], in the context of chronic respiratory disease.

The recruitment and timing of T-cell-mediated immunity is critical for an effective response to viral infection, but this response needs to be tightly regulated to minimize excessive, persistent tissue damage. Indeed, higher CD8:CD4 T cell ratios were previously associated with acute respiratory distress syndrome following respiratory viral infection, most likely due to damage caused by CD8+ cytotoxic T cells [68]. This CD8:CD4 T cell ratio was normalized at 7 dpi in TLR7 KO mice, indicating minimal lung dysfunction at that stage, but it was significantly higher at 14 dpi, consistent with increased hyperresponsiveness. An association between CD8+ T cells and airway hyperreactivity was identified in those mice. Furthermore, Treg frequencies had returned to baseline unlike in WT mice, where Treg levels remained elevated. This reduction in Treg frequencies in TLR7 KO mice could potentially enhance T cell persistence. It was previously shown that the direct stimulation of TLR7 in Tregs can enhance their suppressive functions [69] and attenuate allergic responses [70]. It is therefore possible for TLR7-deficient Tregs in the resolution phase of our model to have a reduced suppressive capacity. Alongside Tregs, Th17 cells can also contribute to virus-induced respiratory dysfunction [71,72]; for instance, the deletion of TLR7 enhanced Th17 cytokine signaling and mucus production following RSV infection [73], potentially enhancing airway hyperactivity and remodeling. Together, these may contribute to persistent T cell-mediated inflammation and influence chronic respiratory dysfunction following IAV infection.

In summary, this study illustrates the important role of TLR7 in exacerbating lung dysfunction during peak IAV infection by intensifying the Th1 inflammatory response in the lungs. Although airway hyperreactivity induced by IAV still ensued in TLR7-deficient mice, it took longer to manifest and was associated with a Th2-driven phenotype. Pharmacologically inhibiting TLR7 during established IAV infection dampened inflammation-induced lung damage in mice [29], suggesting that an early to mid-stage therapeutic suppression of TLR7 could protect against acute inflammation-dependent lung dysfunction. However, our data suggest that chronic TLR7 suppression during acute infection might lead to a shift towards a heightened Th2 response and suppressed Treg response, potentially increasing airway hyperreactivity and allergen sensitivity. Thus, TLR7 is a potential target for therapeutic intervention during IAV infection, but its important time-dependent role must be considered. The significance of TLR7 in establishing adaptive immunological memory following infection also needs consideration, as previously reported [19]. To address this, future studies should investigate the effects of the pharmacological targeting of TLR7 compared to genetically deficient models. This dual role of TLR7 necessitates a stage-dependent, multi-target therapeutic strategy: using a TLR7 antagonist to mitigate inflammation during peak infection, and later employing strategies to counteract Th2 responses such as the TLR7 agonist or anti-IL5 mAb to prevent chronic airway hyperreactivity. More research is required to optimize TLR7-targeted therapies for IAV infection to minimize the overall impact on respiratory health.

## 4. Materials and Methods

### 4.1. Mice and Virus Infection

Male C57BL/6J mice were obtained from the Animal Resources Center (Perth, WA, Australia). Homozygous TLR7 knockout mice (B6.129S1-Tlr7tm1Flv/J) were obtained from the Jackson Laboratory (Bar Harbor, ME, USA) and bred in-house at the RMIT University animal research facility (Bundoora, VIC, Australia). Mice were housed in a 12 h light/12 h dark cycle with food and water.

For infection, 14–18-week-old mice were anesthetized by isoflurane inhalation and inoculated intranasally with PBS control or 50 plaque-forming units (PFUs) of PR8/A virus in a 35 µL volume. Mice were weighed and monitored daily. Mice were euthanized by injection (i.p) of a mixture of ketamine (180 mg/kg) and xylazine (32 mg/kg) on days 7 and 14 of the experimental endpoint. Male mice were used in this study as they mount a more vigorous immune response to IAV infection than female mice [74]. All animal experiments were conducted according to approval obtained from the RMIT University Animal Ethics Committee (Ethics number 23328) and in compliance with the guidelines of the National Health and Medical Research Council of Australia on animal experimentation.

### 4.2. Viral Load by qPCR

Total RNA was extracted from lung homogenates using the RNeasy Mini kit (Qiagen, Germantown, MD, USA), and 2 µg of RNA was converted to cDNA using the High-Capacity cDNA Reverse Transcription Kit (Applied Biosystems, Foster City, CA, USA) according to manufacturer’s instructions. Expression of influenza A polymerase gene was evaluated using a custom-designed TaqMan primer containing the oligonucleotide sequences 5′-CGGTCCAAATTCCTGCTGA-3′ and 5′-CATTGGGTTCCTTCCATCCA-3′, and the TaqMan Fast Advanced Master Mix (ThermoFisher, Scoresby, VIC, Australia). Amplification was performed using a QuantStudio 7 Flex Real-Time PCR system (Thermofisher) according to the following program: 50 °C for 2 min and 95 °C for 2 min, and then 40 cycles of 95 °C for 1 s and 60 °C for 20 s. Gene quantitation was performed in triplicate and normalized against the housekeeping gene RPS18 using the formula 2^−ΔCt^.

### 4.3. Immunophenotyping by Flow Cytometry

Whole lungs (following bronchoalveolar lavage) were finely minced using scissors and then enzymatically digested using 1% Liberase (Sigma-Aldrich, Bayswater, VIC, Australia) for 45 min at 37 °C with shaking at 700 rpm. Tissues were homogenized, and then single-cell suspensions were prepared by straining through a 40 µM strainer. After lysing the red blood cells with ACK lysis buffer, cells were stained with cocktail mixtures of fluorescent-labeled anti-mouse antibodies diluted in FACS buffer (PBS + 2.5% FBS) for 30 min on ice. The following Biolegend (San Diego, CA, USA) antibodies were used (unless stated otherwise): CD45-Alexa Fluor 700 (clone 30-F11), CD45-BV650 (clone 30-F11), CD4-APC-Cy7 (clone GK1.5), CD8a-PerCP (clone 53-6.7), NK1.1-BV605 (clone PK136), CD11b-BV421 (clone M1/70), CD11b-APC-Cy7 (clone M1/70), CD11c-PE-Cy7 (clone N418; eBioscience, Waltham, MA, USA), CD69-BV650 (clone H1.2F3), PDCA-1-Pacific Blue (clone 217108), MHC-II-APC (clone M5/114.15.2), Ly6C-PerCP (clone HK1.4), Ly6G-APC-Cy7 (clone 1A8), Siglet-F-PE (clone 1RNM44N; ThermoFisher), and F4/80-PE (clone BM8; eBiocience). Cells were fixed and permeabilized using the eBioscience Foxp3/Transcription Factor Staining Buffer Set (ThermoFisher) for intracellular staining with FoxP3-BV421 (clone MF-14). CD16/32 (clone 2.4G2) and LIVE/DEAD Fixable Aqua Dead Cell Stain Kit (Invitrogen, Carlsbad, CA, USA) were contained within each antibody cocktail mixture to block Fc-mediated adherence of the antibodies and to exclude dead cells, respectively. Samples were processed on a BD LSRFortessa X-20 flow cytometry analyzer with DIVA v9.0 software (Becton Dickinson Bioscience, Macquarie Park, NSW, Australia) and data were analyzed using FlowJo v10 software (Tree Star, Inc., Ashland, OR, USA). The representative gating strategy is shown in Appendix A.

### 4.4. Cytokine Protein Expression

Secreted cytokine and chemokine expression was measured using an 8-Plex Mouse Luminex Discovery Assay kit (R&D Systems, Minneapolis, MN, USA, cat #LXSAMSM-08). Briefly, 50 µL of bronchoalveolar lavage fluid (BALF) was added in duplicate, and incubations were performed according to manufacturer’s instructions. Plates were read on a Bio-Plex 200 instrument (Bio-Rad, Gladesville, NSW, Australia) at 50 counts per region and median fluorescence intensity measured. Quantification of IL-4, IL-5 or CCL11 was performed separately using Mouse DuoSet ELISA kits (R&D Systems). Fifty microliters of BALF was added in duplicate to pre-coated 96-well plate and incubations were performed according to manufacturer’s instructions. The plates were read on the CLARIOstar (BMG, Mornington, VIC, Australia) at a wavelength of 450 nm. Cytokine titers in the samples were determined by plotting the optical densities, using a four-parameter fit for the standard curve and expressed in pg/mL.

### 4.5. Lung Function Analysis

Mice were anesthetized with ketamine (80 mg/kg) and xylazine (16 mg/kg), after which tracheotomy was performed by inserting an 18-gauge canular into the trachea. Mice were then mechanically ventilated at 150 breaths per minute using the Flexivent FX1 system (SCIREQ, Montréal, QC, Canada). Forced oscillation technique was used to measure pressure, flow, and volume responses at baseline or in response to 100 mg/mL of nebulized methacholine (MCh; Sigma-Aldrich). Total respiratory system resistance (Rrs, representing resistance of lung tissue and conducting airways) and respiratory system elastance (Ers, representing elastic properties of lung tissue and chest wall) was measured using the single-compartment model. Newtonian resistance (Rn, representing resistance of conducting airways), tissue damping (G, representing energy dissipation in alveoli), and tissue elastance (H, representing energy conservation in alveoli) were measured using the constant phase model.

### 4.6. Statistical Analysis

All data are expressed as the mean ± SEM from at least two independent experiments. All comparisons were performed using GraphPad Prism (GraphPad Software Version 10.1.2, Boston, MA, USA) and performed by two-way ANOVA followed by Tukey’s post hoc tests for multiple comparisons. Simple linear regression analysis was also performed on some datasets. Statistical significance was considered at *p* < 0.05.

## Figures and Tables

**Figure 1 ijms-25-13699-f001:**
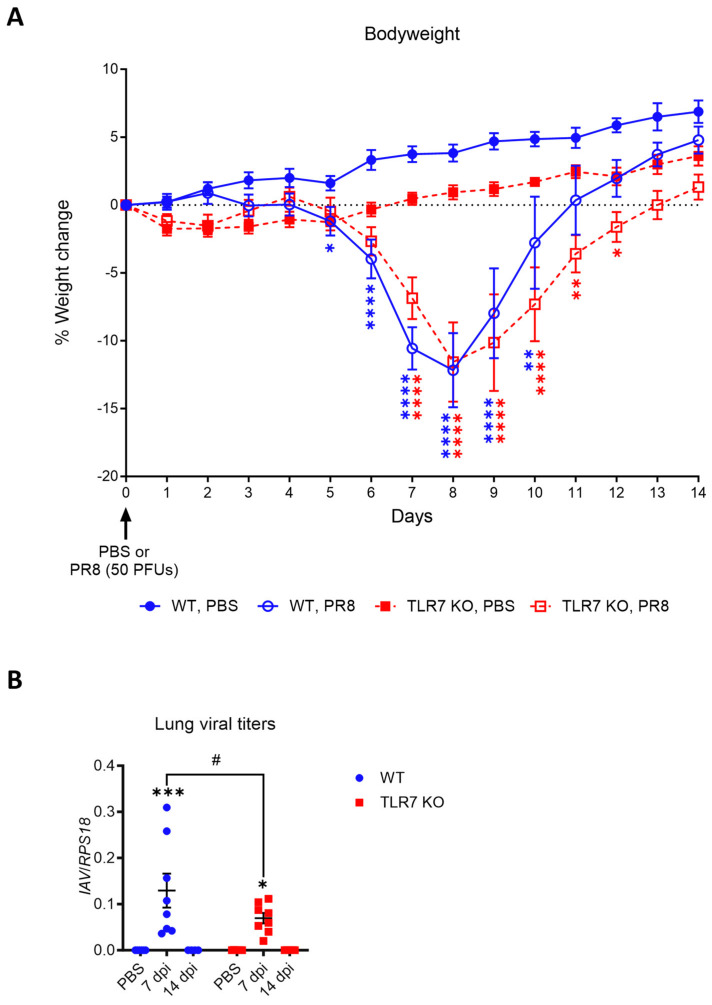
PR8-induced bodyweight loss is slightly delayed in TLR7 KO mice. WT C57Bl/6 or TLR7 KO mice were infected with PR8 (50 PFUs) or PBS (control). (**A**) Bodyweights were recorded daily for 14 days and presented as % weight change from day of infection. (**B**) Influenza A virus polymerase mRNA expression in the lungs was measured by RT-qPCR and is presented relative to RPS18 housekeeping (2^−ΔCt^). Data are expressed as mean ± SEM, *n* = 6–12 pooled from two to three independent experiments. Statistical analysis was conducted using a two-way ANOVA test followed by Tukey’s post hoc test for multiple comparisons (* *p* < 0.05, ** *p* < 0.01, *** *p* < 0.001, **** *p* < 0.0001 for comparison of each infected genotype at 7 or 14 dpi with relative uninfected group; # *p* < 0.05 for comparison between genotypes for a specific timepoint).

**Figure 2 ijms-25-13699-f002:**
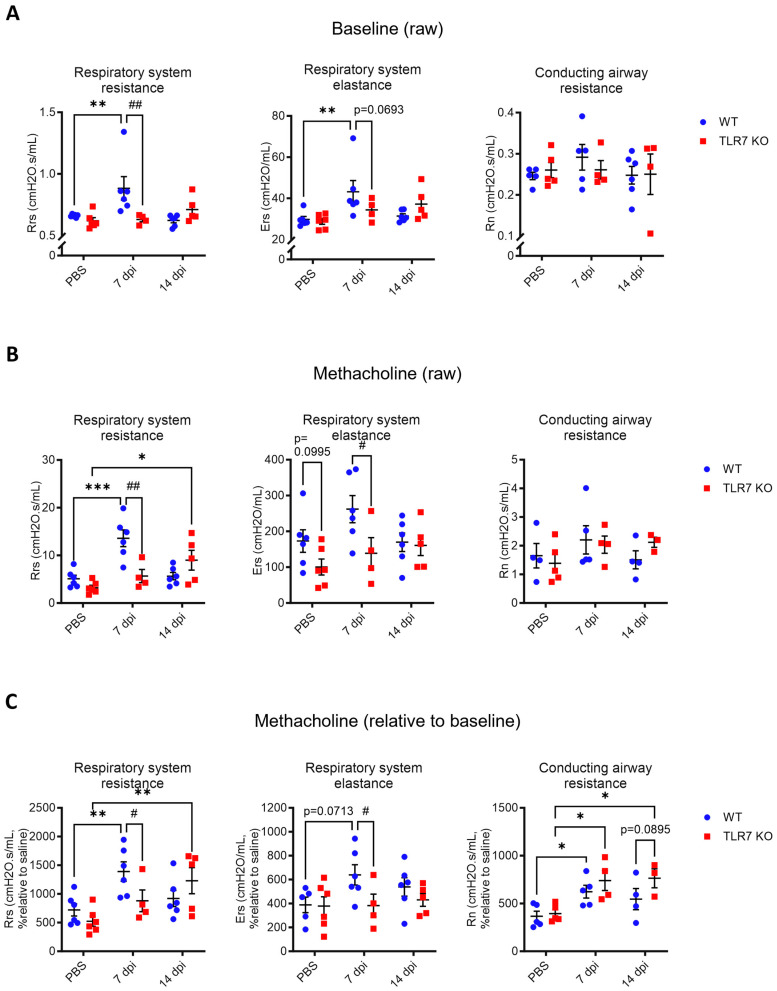
PR8-induced airway hyperresponsiveness is delayed in TLR7 KO mice. WT C57Bl/6 or TLR7 KO mice were infected with PR8 (50 PFUs) or PBS (control). Mice were anesthetized after 7 or 14 dpi and mechanically ventilated at 150 breaths per minute using the Flexivent FX1 system. Airway hyperreactivity was determined in response to nebulized methacholine (MCh; 0 or 100 mg/mL). Maximal respiratory system resistance (Rrs), respiratory system elastance (Ers), and conducting airway resistance (Rn) were measured. Data are presented as (**A**) baseline raw values and (**B**) raw or (**C**) relative values (fold change over baseline) after MCh exposure. Data are expressed as mean ± SEM, *n* = 4–6 mice per experimental group pooled from two independent experiments. Statistical analysis was conducted using two-way ANOVA test followed by Tukey’s post hoc test for multiple comparisons (* *p* < 0.05, ** *p* < 0.01, *** *p* < 0.001 for comparison of each infected genotype at 7 or 14 dpi with relative uninfected group; # *p* < 0.05, ## *p* < 0.01 for comparison between genotypes for a specific timepoint).

**Figure 3 ijms-25-13699-f003:**
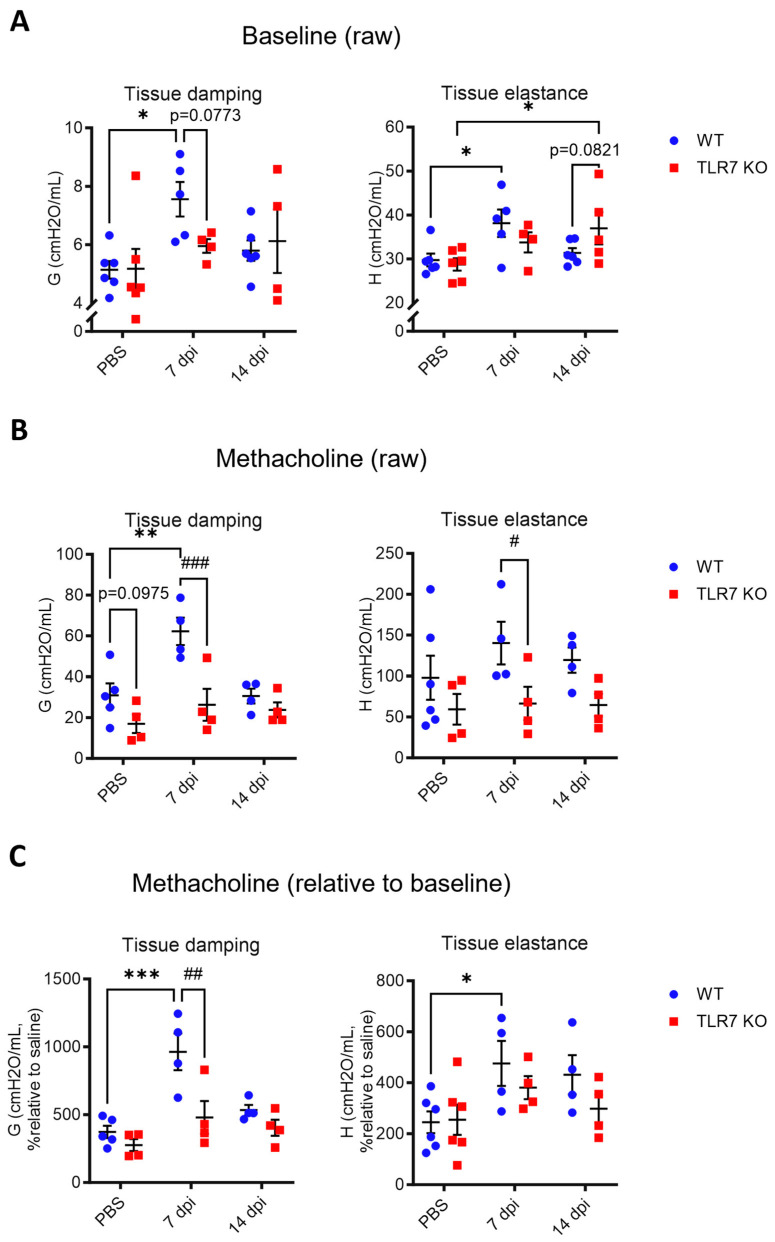
PR8-induced lung tissue dysfunction is spared in TLR7 KO mice. WT C57Bl/6 or TLR7 KO mice were infected with PR8 (50 PFUs) or PBS (control). Mice were anesthetized after 7 or 14 dpi and mechanically ventilated at 150 breaths per minute using the Flexivent FX1 system. Tissue hyperreactivity was determined in response to nebulized methacholine (MCh; 0 or 100 mg/mL). Maximal tissue damping (G) and tissue elastance (H) were measured. Data are presented as (**A**) baseline raw values and (**B**) raw or (**C**) relative values (fold change over baseline) after MCh exposure. Data are expressed as mean ± SEM, *n* = 4–6 mice per experimental group pooled from two independent experiments. Statistical analysis was conducted using two-way ANOVA test followed by Tukey’s post hoc test for multiple comparisons (* *p* < 0.05, ** *p* < 0.01, *** *p* < 0.001 for comparison of each infected genotype at 7 or 14 dpi with the relative uninfected group; # *p* < 0.05, ## *p* < 0.01, ### *p* < 0.001 for comparison between genotypes for a specific timepoint).

**Figure 4 ijms-25-13699-f004:**
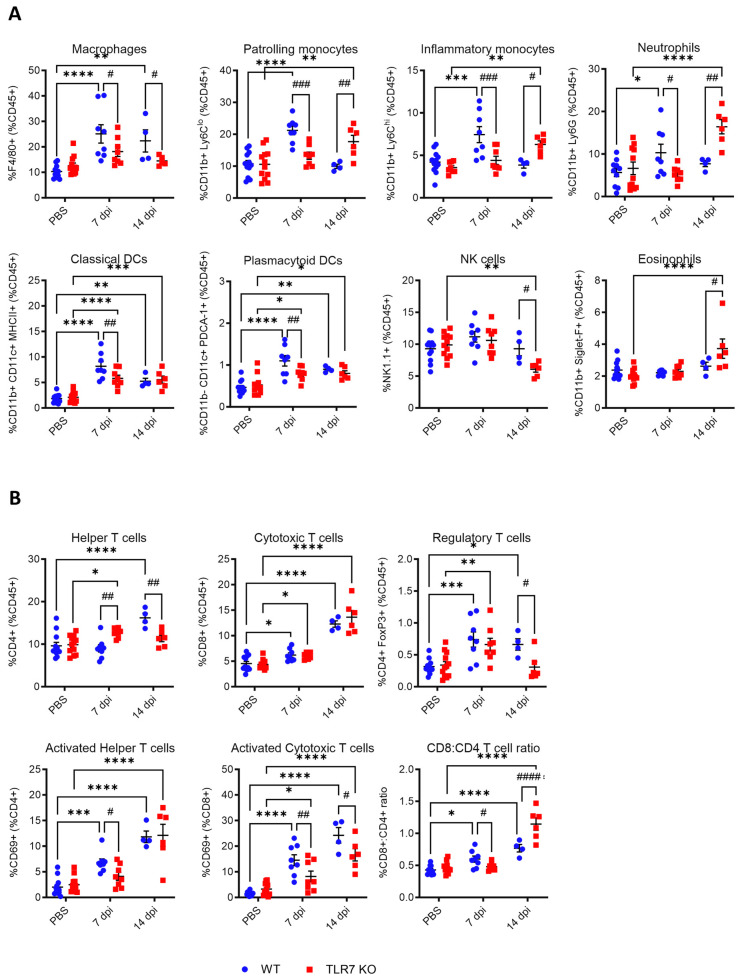
Immune cell recruitment in the lung tissue is altered in TLR7 KO mice across PR8 infection. WT C57Bl/6 or TLR7 KO mice were infected with PR8 (50 PFUs) or PBS (control) and sacrificed after 7 or 14 days. Immune cell populations in the lung tissue were determined using flow cytometry. (**A**) Innate immune cell types: macrophages (F4/80+), patrolling monocytes (CD11b+ Ly6C^lo^), inflammatory monocytes (CD11b+ Ly6C^hi^), neutrophils (CD11b+ Ly6G+), classic DCs (CD11b+ CD11c+ MHCII+), plasmacytoid DCs (CD11b- CD11c+ PDCA-1+), NK cells (NK1.1+), and eosinophils (CD11b+ Siglet F+). (**B**) Adaptive T cell types: T helper (CD3+ CD4+) and cytotoxic T cells (CD3+ CD8+), regulatory T cells (CD4+ FoxP3+), and activated T cell subsets (CD69+). Cell frequencies were measured as a proportion of the CD45+ population. Data are expressed as mean ± SEM, *n* = 6-8 mice per experimental group pooled from two independent experiments. Statistical analysis was conducted using two-way ANOVA test followed by Tukey’s post hoc test for multiple comparisons (* *p* < 0.05, ** *p* < 0.01, *** *p* < 0.001, **** *p* < 0.0001 for comparison of each infected genotype at 7 or 14 dpi with relatively uninfected group; # *p* < 0.05, ## *p* < 0.01, ### *p* < 0.001, #### *p* < 0.0001 for comparison between genotypes for a specific timepoint).

**Figure 5 ijms-25-13699-f005:**
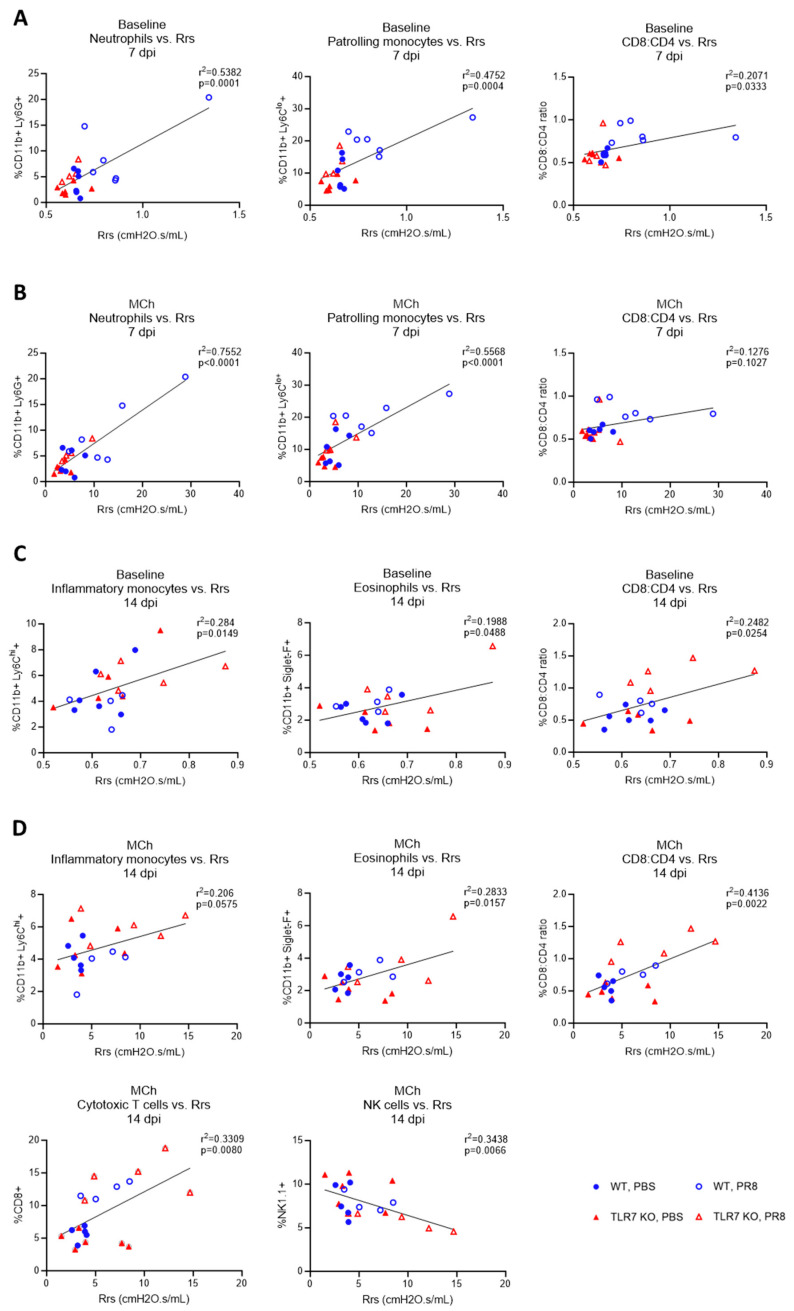
Immune cell correlations with airway hyperresponsiveness across PR8 infection. Simple linear regression tests were performed on C57Bl/6 or TLR7 KO mice that were infected with PR8 (50 PFUs) or PBS (control) after 7 or 14 dpi. For each mouse, comparisons were made for lung-infiltrating immune cell types with the degree of respiratory system resistance (Rrs). Analysis at (**A**) baseline or (**B**) in response to 100 mg/mL nebulized methacholine (MCh) after 7 dpi, or (**C**) at baseline or (**D**) MCh after 14 dpi, is presented. Data are expressed as mean ± SEM, *n* = 4–6 mice per experimental group pooled from two independent experiments. Statistical analysis was conducted using a simple linear regression test.

**Figure 6 ijms-25-13699-f006:**
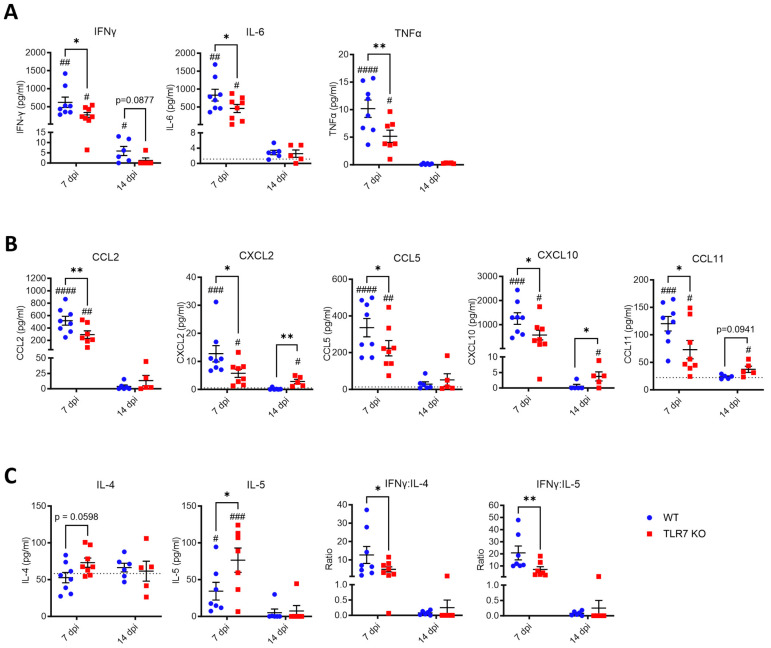
Reduced inflammatory cytokine markers in the lungs of TLR7 KO mice following acute PR8 infection. WT C57Bl/6 or TLR7 KO mice were infected with PR8 (50 PFUs) or PBS (control) bronchoalveolar lavage (BAL) performed on terminal mice after 7 or 14 days. Protein expression of (**A**) inflammatory cytokines, (**B**) chemokines, and (**C**) Th2 cytokines in the BAL fluid were measured by immunoassays. Data from infected mice are presented and expressed as mean ± SEM, *n* = 6–8 mice per experimental group pooled from two independent experiments. The dotted line indicates the total mean protein values from uninfected groups. Statistical analysis was conducted using a two-way ANOVA test followed by Tukey’s post hoc test for multiple comparisons (* *p* < 0.05, ** *p* < 0.01 for comparison between genotypes; # *p* < 0.05, ## *p* < 0.01, ### *p* < 0.001, #### *p* < 0.0001 for comparison of each genotype relative to uninfected controls).

**Table 1 ijms-25-13699-t001:** Summary of linear regression analysis of Rrs at baseline and MCh-induced with lung immune cell populations across PR8 infection. Red font indicates the significant or near-significant values.

Correlation with Rrs	7 dpi				14 dpi			
	Baseline		Methacholine		Baseline		Methacholine	
Cell Type	*p*-Value	R^2^	*p*-Value	R^2^	*p*-Value	R^2^	*p*-Value	R^2^
Macrophages	0.1738	0.0904	0.4293	0.0315	0.4213	0.0408	0.3042	0.0658
Inflammatory monocytes	0.2834	0.0573	0.6429	0.0109	0.0149	0.287	0.0575	0.206
Patrolling monocytes	0.0004	0.4752	<0.0001	0.5568	0.6627	0.0108	0.4235	0.0358
Neutrophils	0.0001	0.5382	<0.0001	0.7552	0.3359	0.0753	0.3003	0.0668
Classic DCs	0.0894	0.1375	0.1196	0.1168	0.8333	0.0025	0.3994	0.0397
Plasmacytoid DCs	0.0555	0.1713	0.1618	0.0954	0.9658	0.0001	0.6739	0.01
Eosinophils	0.4653	0.0269	0.6612	0.00097	0.0488	0.1988	0.0157	0.2833
NK cells	0.416	0.0333	0.1135	0.109	0.0678	−0.1734	0.0066	−0.3438
Regulatory T cells	0.315	0.0504	0.4407	0.03	0.0391	−0.2156	0.7591	−0.0053
Helper T cells	0.2519	−0.06509	0.1192	−0.117	0.2682	−0.0067	0.7375	0.0064
Cytotoxic T cells	0.5716	0.01625	0.8548	0.0017	0.2769	0.0653	0.008	0.3309
CD8:CD4 ratio	0.0333	0.2071	0.1027	0.1276	0.0254	0.2482	0.0022	0.4136

## Data Availability

The data presented in this study are available in this article and the relevant Appendix A.

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
