# Peer review of "TLR7 Promotes Acute Inflammatory-Driven Lung Dysfunction in Influenza-Infected Mice but Prevents Late Airway Hyperresponsiveness"

_ijms, 2024, doi:10.3390/ijms252413699_

Round 1
Reviewer 1 Report
Comments and Suggestions for Authors
Dear Author,
Miles et al. present the results of a study into the contribution of TLR7 to the pathology of influenza in a mouse model.
Wild type and TLR7 k/o mice received PR8 strain influenza virus and were monitored for for 14 days with harvests at day 7 and 14. TLR7 k/o mice exhibited initially delayed but then exaggerated body weight loss. Somewhat paradoxically, the TLR7 k/o mice had lower viral titres at day 7, with both genetic types fully clearing the infection by day 14. In WT mice, airway resistance and elastance were increased at day 7 post-infection and returned to normal at day 14, but there was no increase in TLR7 k/o mice. When hyperresponse was induced with methacholine, the increased resistance persisted to the day 14 timepoint. Tissue damping and elastance were similarly increased at day 7 of infection and returned to baseline at day 14 but not in the TLR7 k/o mice. This pattern was also observed after methacholine administration to model an asthma attach.
Single cell suspension was generated from the lungs which showed the expected infiltration of leukocytes at day 7 post-infection which had almost returned to normal at day 14. In TLR7 k/o mice, macrophage infiltration had normal temporal dynamics but other monocytes and neutrophils had a very delayed infiltration profile. For T-cell population in the lung, helper T-cells arrive earlier but take longer to abate in the TLR7 k/o, cytotoxic T-cells are unchanged and regulatory T-cells abate more rapidly. This shows a very nuanced change in infiltration and clearance reliant on TLR7 signalling possible favouring a less inflammatory Th2 response or at least a delayed inflammatory response.
The cytology findings were largely reflected by the cytokine analysis of the BAL. Pro-inflammatory cytokines were reduced at day 7 in TLR7 k/o mice, but were somewhat elevated at day 14. IL-4 and IL-5 level were conversely increased at day 7 in the TLR7 k/o animals.
This experimental techniques are appropriate and appear to be competently performed. The standard of English is good throughout and it is easy to read.
Major comments
--------------
1. The rate of weight gain actually appears to be identical in the wild type and knockout mice, indicating the starting weight must be different? Is there an explanation for this.
2. Has the group ever performed a more detailed viral load study to determine that day 7 has the maximum viral load and other injury parameters?
3. I don't understand the presentation of Figure 5. Why aren't there two slope lines in each graph comparing WT and k/o?
4. Was tissue histology performed? (Perhaps all of the tissue went to the single cell suspension).
Minor comments
--------------
P1
"overstimulation of host inflammation"
> I would avoid this wording. For 99% of infections the stimulation is entirely correct and results in pathogen clearance.
P8
Figure 3A presents data in two different ways, making interpretation more difficult.
Regards
Author Response
Major comments
Comment 1: The rate of weight gain actually appears to be identical in the wild type and knockout mice, indicating the starting weight must be different? Is there an explanation for this.
Author response 1: While the mice used in the experiment were of similar age, the TLR7 KO mice were on average 1 gram heavier than the WT mice at the start. To account for this slight difference in starting body weight, we presented the bodyweight trends across infection relative to starting weight to identify infection-induced effects. Statistical comparisons for this dataset were then made only between uninfected and infected groups of each genotype, and not between genotypes.
Comment 2: Has the group ever performed a more detailed viral load study to determine that day 7 has the maximum viral load and other injury parameters?
Author response 2: Thank you for raising this point. We have not performed additional viral load analysis on the tissue used in this study. The kinetics of PR8 infection in mice have been reported previously (PMC8370774) where peak viral load in the lungs following intranasal inoculation occurs between 3-6 dpi, followed by a decline after day 7 and clearance by days 9-10. We therefore believe that day 7 represents a timepoint of acute infection with high viral load in the lung.
We have now included this citation in line 118 to acknowledge that “These timepoints coincide with peak inflammation and viral clearance [new ref 30]…”
Comment 3: I don't understand the presentation of Figure 5. Why aren't there two slope lines in each graph comparing WT and k/o?
Author response 3: We recognize the potential confusion for Fig 5. Our intention in presenting a single slope for all 4 groups was to identify any correlations across the entire dataset, encompassing both genotypes. This approach provides a clearer understanding of how immune cells may correlate with lung dysfunction across all groups at a particular timepoint. As such, we used distinct symbols for each mouse (genotype and infection group) on the graph to allow for the visualization of any group-specific trends within the analysis.
Comment 4: Was tissue histology performed? (Perhaps all of the tissue went to the single cell suspension).
Author response 4: Thank you for raising this point. We have not been able to perform histological analysis on the lungs from the mice used in this study as the tissue was used for flow cytometric analysis.
Minor comments
P1 "overstimulation of host inflammation" > I would avoid this wording. For 99% of infections the stimulation is entirely correct and results in pathogen clearance.
Author response: We reworded this sentence to “Therefore, the hyperactivation of host inflammatory responses by IAV can play a critical role in driving both acute and chronic respiratory complications following infection”
P8 Figure 3A presents data in two different ways, making interpretation more difficult.
Author response: Thank you for bringing this to our attention as it was our error. We have now fixed the style of the graph in Fig 3A to be consistent with the remaining graphs.
Reviewer 2 Report
Comments and Suggestions for Authors
In their article “TLR7 Promotes Acute Inflammatory Lung Dysfunction in Influenza-Infected Mice but Prevents Late Airway Hyperresponsiveness,” the authors present a study on the effect of toll-like receptor 7 on lung function during peak influenza infection.
Severe respiratory illness often leads to serious pulmonary complications, which can lead to excessive lung inflammation due to a “cytokine storm.” According to the study, activation of TLR7 early in infection can enhance early antiviral IFN responses to protect against the virus, while inhibition of TLR7 after infection is established can suppress the cytokine storm and inflammatory lung damage.
The topic is very interesting and important, due to the seasonal influenza epidemic and the constant threat of a new pandemic.
The manuscript is well written, I have only a few comments:
For a better visualization, it is best to follow the following section order: introduction, materials and methods, results, discussion. The ethics statement is usually at the end of the article.
Results. Figures should have a title. After that, the text is quite detailed and the statistical method you used is repeated throughout, it might be better to just note the p-values.
Lines 157-160, 247-254, 286-288 - Comments are better suited to the Discussion section.
Discussion.
Line 348 - “resiquimod” should be uppercase.
Author Response
The manuscript is well written, I have only a few comments:
Comment 1: For a better visualization, it is best to follow the following section order: introduction, materials and methods, results, discussion. The ethics statement is usually at the end of the article.
Author response 1: We have formatted the manuscript in accordance with the IJMS article template. As such we have not altered the current format, unless requested by the editor.
Comment 2: Results. Figures should have a title. After that, the text is quite detailed and the statistical method you used is repeated throughout, it might be better to just note the p-values.
Author response 2: We thank the reviewer for this suggestion. In accordance with the IJMS article template, the first line of each figure caption is considered the title of the figure. While we used consistent statistical methods, the presentation of each figure differs, and we feel it is clearer to specify each figure in appropriate detail alongside the statistical comparisons applied to that figure. We can update this if the editor feels it is necessary.
Comment 3: Lines 157-160, 247-254, 286-288 - Comments are better suited to the Discussion section.
Author response 3: Thank you for this suggestion. We have carefully considered your recommendation to move the indicated comments from the results section to the discussion. However, we believe that these comments are essential for objectively summarizing the data in the results section before providing further interpretation in the discussion. Therefore, we have decided to retain their current placement, as it helps maintain the clarity and flow of the manuscript. We can make revisions if the editor feels it is necessary.
Comment 4: Discussion. Line 348 - “resiquimod” should be uppercase.
Author response 4: We have made this change as suggested.
Reviewer 3 Report
Comments and Suggestions for Authors
The paper is interesting and well written. The authors investigated how TLR7-related inflammation affects lung function following influenza A virus infection. The study highlighted an underappreciated duality of TLR7 in IAV disease: promoting inflammation-driven lung dysfunction during the acute infection
but suppressing eosinophilic and CD8+ T cell dependent hyperresponsiveness during disease resolution. The methodology is adequate and well described. The statistical analysis is coerent. The results and discussion are adequate and well described. I suggest to discuss the role of Th17 cells in chronic inflammatory immune mediated diseases, and the role of IL-31/IL-33 axis, vitamin D and microbioma in immune responses (see and add as references papers by Murdaca et al concerning Th17 cells in chronic inflammatory immune mediated diseases, IL-31/IL-33 axis, vitamin D and microbioma in immune responses).
Minor english editing
Author Response
Comment: I suggest to discuss the role of Th17 cells in chronic inflammatory immune mediated diseases, and the role of IL-31/IL-33 axis, vitamin D and microbioma in immune responses (see and add as references papers by Murdaca et al concerning Th17 cells in chronic inflammatory immune mediated diseases, IL-31/IL-33 axis, vitamin D and microbioma in immune responses).
Author response: Thank you for suggesting these additional discussion points. We have now included additional text in the discussion addressing these areas and the potential link with TLR7.
Lines 400-403: “Further research could also consider the effect of TLR7 on other pathways involved in the amplification of the Th2 response, such as IL-13, IL-31 and IL-33 cytokines, vitamin D deficiency and changes in the lung microbiome [new refs 66,67], in the context of chronic respiratory disease.”
Lines 417-419: “Alongside Tregs, Th17 cells can also contribute to virus-induced respiratory dysfunction [new refs 71,72]. For instance, the deletion of TLR7 enhanced Th17 cytokine signaling and mucus production following RSV infection [new refs 73], potentially enhancing airway hyperactivity and remodeling.”
Reviewer 4 Report
Comments and Suggestions for Authors
In the current study by Miles et al., the authors worked on assessing the role of TLR7 in the context of IAV infection and reported a delayed innate inflammatory response to PR8 infection in the absence of TLR7. These observations were complemented by a reduced numbers of activated T cells, delayed influx of inflammatory myeloid cells to the lungs, reduced levels of Th1 cytokines IFNγ, IL-6 and TNFα in the BALF and reduced titers of chemokines in TLR7 deficient mice. Finally, they reported that TLR7 deficiency lessened lung inflammation and suppressed the onset of lung hyperresponsiveness during acute infection.
This manuscript can be benefited from addressing the following points.
· The authors reported that loss of TLR7 dampened the innate immune response in the acute phase but increased neutrophils and Ly6Clo and Ly6Chi monocytes at 14 dpi compared to uninfected controls. This suggests a delayed innate inflammatory response to PR8 infection in the absence of TLR7 but not a dual action of TLR7. On these lines, the title is a mislead to the readers and needs authors.
· Increased titers of IL-5 were reported in TLR7KO mice at 14dpi compared to WT and recited enhanced Th2 signaling while no changes were seen with IL-4 and IL-13 whose responses help to eliminate infections. It would be ideal to show IL-13 data as well. Moreover, as loss of TLR7 resulted in dampening the infection burden in the acute phase, the observations seem in late phase of infection are the early onset of damage repair. This highlights the role of TLR7 in acute infection phase with no major role in late stage of infection.
· PMID 22837197 reported that “TLR7 recognition is dispensable for IAV Infection but important for the induction of Hemagglutinin-Specific Antibodies in response to the 2009 Pandemic Split Vaccine in Mice”. On these lines, do the authors believe in using a TLR7 antagonist to mitigate inflammation during peak infection, and later employing strategies to counteract Th2 responses such as TLR7 agonist or anti-IL5 mAb prevent chronic airway hyperreactivity, as stated in their discussion?
· As the manuscript was focused detailing the immune profile post IAV infection and innate inflammatory conditions while emphasizing the role of TLR7, it would be more interesting to see day 3 post infection data, in the context of innate immune cell profile!
Author Response
This manuscript can be benefited from addressing the following points.
Comment 1: The authors reported that loss of TLR7 dampened the innate immune response in the acute phase but increased neutrophils and Ly6Clo and Ly6Chi monocytes at 14 dpi compared to uninfected controls. This suggests a delayed innate inflammatory response to PR8 infection in the absence of TLR7 but not a dual action of TLR7. On these lines, the title is a mislead to the readers and needs authors.
Author response 1: We thank this reviewer for raising this point. We have acknowledged that our data suggests a delayed innate inflammatory response to infection (lines 187-188). However the findings that infected mice exhibited Th2 signaling, as evidenced by eosinophilia and IL-5 production, only upon TLR7 deficiency suggests that this is a phenotypic effect rather than a delay in the overall innate response. We discuss previous literature supporting a Th1 dominant, Th2 suppressing role of TLR7 in the context of infection or allergic/asthma disease, to further support those findings (lines 378-382, 386-387).
Comment 2: Increased titers of IL-5 were reported in TLR7KO mice at 14dpi compared to WT and recited enhanced Th2 signaling while no changes were seen with IL-4 and IL-13 whose responses help to eliminate infections. It would be ideal to show IL-13 data as well. Moreover, as loss of TLR7 resulted in dampening the infection burden in the acute phase, the observations seem in late phase of infection are the early onset of damage repair. This highlights the role of TLR7 in acute infection phase with no major role in late stage of infection.
Author response 2: We thank this reviewer for raising this point. We acknowledge that IL-13 is an important Th2 cytokine involved in viral infection. The Luminex assays and ELISAs employed in this study did not measure IL-13 levels. Therefore, we have suggested in the discussion that IL-13 signaling should be considered in future work to further explore the Th2 response following IAV infection.
Lines 400-403: “Further research could also consider the effect of TLR7 on other pathways involved in the amplification of the Th2 response, such as IL-13, IL-31 and IL-33 cytokines, vitamin D deficiency and changes in the lung microbiome [65,66], in the context of chronic respiratory disease.”
Comment 3: PMID 22837197 reported that “TLR7 recognition is dispensable for IAV Infection but important for the induction of Hemagglutinin-Specific Antibodies in response to the 2009 Pandemic Split Vaccine in Mice”. On these lines, do the authors believe in using a TLR7 antagonist to mitigate inflammation during peak infection, and later employing strategies to counteract Th2 responses such as TLR7 agonist or anti-IL5 mAb prevent chronic airway hyperreactivity, as stated in their discussion?
Author response 3: Thank you for raising this very intriguing point. We have considered this idea in our scientific discussions and may not have appropriately addressed it in the discussion aside from acknowledging that particular study in the introduction. We have therefore added extra text to the discussion to acknowledge this.
Lines 434-435: “The significance of TLR7 in establishing adaptive immunological memory following infection also needs consideration as previously reported [19].”
Comment 4: As the manuscript was focused detailing the immune profile post IAV infection and innate inflammatory conditions while emphasizing the role of TLR7, it would be more interesting to see day 3 post infection data, in the context of innate immune cell profile!
Author response 4: We agree with this important point and believe that examining the early innate response during early IAV infection in TLR7-deficient mice could provide valuable insights. These early timepoints in TLR7 KO mice following IAV infection have already been reported in other studies (PMC5819576 - ref 55, PMC21966467 – ref 16). The focus and novelty of our study lie in our analysis of the later stages of infection and how changes in the lung immune profile at this stage can influence lung function. We feel that additional analysis at earlier stages of infection may shift the focus away from the primary objective of this particular study.